# High-impedance microwave resonators with two-photon nonlinear effects

S. Andersson ⓘ, H. Havir ⓘ, A. Ranni ⓘ, S. Haldar ⓘ & V. F. Maisi ⓘ ✉

Nonlinear effects play a central role in photonics as they form the foundation for most of the device functionalities such as amplification and quantum state preparation and detection. Typically the nonlinear effects are weak and emerge only at high photon numbers with strong drive. Here we present an experimental study of a Josephson junction -based high-impedance resonator. We show that by taking the resonator to the limit of consisting effectively only of one junction, results in strong nonlinear effects already for the second photon while maintaining a high impedance of the resonance mode. Our experiment yields thus resonators with strong interactions both between individual resonator photons and from the resonator photons to other electric quantum systems. We also present an energy diagram technique which enables to measure, identify and analyse different multi-photon optics processes along their energy conservation lines.

Microwave photons residing in electric resonators find use in a wide range of systems. They are the standard approach to obtain long distance coherent coupling and readout for quantum devices ranging from spin to superconducting systems, and form for example bosonic qubits inside the resonators. In recent years, high-impedance resonators[1–11] have gained much attention as they increase the zero-point fluctuations of the electric field to reach strong coupling regime for charge[5,12,13] and spin qubits[7,14–17]. On the other hand, nonlinear effects - which are typically weak in the few photon limit - are interesting as they provide photonic interactions, for example, for photon routing[18,19], parametric amplification[20–22], electrostatically tunable fluorescence[23,24] and quantum state preparation[25,26]. Recently, nonlinear effects taking place in a microwave resonator at the few photon limit was achieved by embedding a nonlinear transmon qubit into a linear low-impedance resonator[26,27]. So far, the few photon nonlinearities have remained untapped for the high impedance systems, despite it would enable to build nonlinear quantum optics devices with unprecedentedly strong and fast interaction dynamics. Here, we show that taking a high-impedance Josephson junction (JJ) array resonator[1–3,28] to the limit of consisting effectively only of one junction achieves strong nonlinear effects already for the second photon while maintaining a high impedance of the resonance mode. Our experiment yields thus resonators with the strong interactions between individual resonator photons, complementing the earlier JJ resonator works[5,13,29–31] coupling the resonator photons strongly to other electric quantum systems.

## Results

Figure 1 a, b present the realization of the high-impedance nonlinear microwave resonator, and Fig. 1c the corresponding electric circuit. The resonance mode with one-photon resonance frequency of $f_{01} \approx 1/2\pi\sqrt{LC_\Sigma}$ forms from a total capacitance $C_\Sigma$ and inductance $L$ of a JJ. Here, the total capacitance $C_\Sigma$ contains the JJ capacitance, capacitance from the resonator middle line (shown in teal) to ground, and two input capacitances $C_c$ which connect the resonator to input and output lines (shown in blue). The single junction geometry yields orders of magnitude stronger nonlinearity as compared to the JJ arrays[5,20,28,32,33] as the full voltage amplitude is across the single junction. We further split the single JJ into a SQUID geometry, Fig. 1b, to tune the value of $L$ with a magnetic field[1,5,20]. One end of the resonator connects to the ground shown in dark gray in Fig. 1a while the other end (in teal) connects to the input and output ports (in blue) via inter-digitized coupling capacitors.

The measured transmission coefficient $T$ is presented in Fig. 1d as a function of the drive frequency $f$ and magnetic flux $\Phi$. Here we apply a drive signal with power $P_1 = 8.6$ aW to the left port and measure the transmitted signal in the right port. The data presents the typical SQUID response: a single resonance mode oscillates periodically

NanoLund and Solid State Physics, Lund University, Box 118, 22100 Lund, Sweden. ✉e-mail: ville.maisi@ftf.lth.se

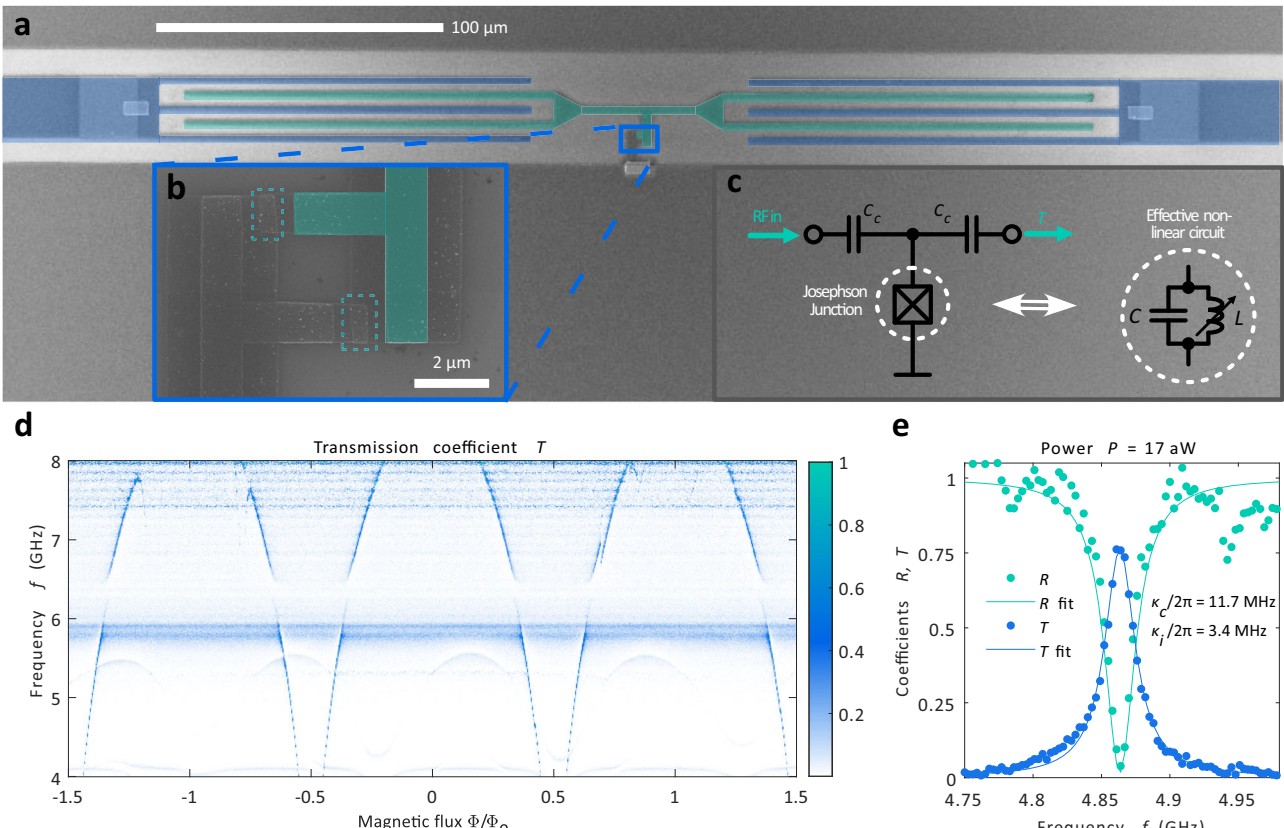

**Fig. 1 | Single Josephson junction resonator. a** A scanning electron micrograph of the measured resonator device. The resonance mode forms across a Josephson junction connecting the bottom ground plane (dark grey) to the resonator middle line (colored in teal). The resonator is driven and measured via an input and output lines colored in blue. **b** A zoom-in to the junction area. The single junction is split into two junctions highlighted with dashed teal rectangles to form a SQUID geometry for tuning the resonance frequency. **c** The effective electric circuit for the first photon. **d** The resonator transmission coefficient $T$ as a function of magnetic flux $\Phi$ and drive frequency $f$. At frequency $f = 4.7$ GHz the input power is $P = 8.6$ aW (see the Methods section for details). **e** Transmission $T$ and reflection coefficient $R$ measured at $\Phi = 0.4\Phi_0$. The solid lines are fits to Eq. (1) of ref. 28 with $f_{01} = 4.86$ GHz, $\kappa_c/2\pi = 12$ MHz and $\kappa_i/2\pi = 3$ MHz.

throughout the 4–8 GHz measurement band of our setup. Figure 1e presents the response at $\Phi = 0.4\Phi_0$ together with the reflection coefficient measured from the left port. The resonator response follows a Lorentzian lineshape with the transmission reaching close to unity, $T = 0.75$, and reflection coefficient close to zero, $R = 0.04$, in resonance. These values are close to the ideal response of the symmetric two port resonator and hence reflect the fact that the internal losses are small compared to the couplings. Fitting to Eq. (1) of ref. 28 yields $f_{01} = 4.86$ GHz, input couplings $\kappa_c/2\pi = 12$ MHz and internal losses of $\kappa_i/2\pi = 3$ MHz.

Figure 2 a–c present the input power dependency of the measured transmission response. In Fig. 2a, we see that the response is constant for low input power up to about $P_1 = 100$ aW. Above this power, we observe a bleaching effect, i.e. the resonator transmission decreases and at around $P_1 = 1000$ aW, the response is suppressed by an order of magnitude, see the linecuts of Fig. 2b. At the highest input power, even the total output power on resonance $f = f_r$ reduces as presented in Fig. 2c.

To compare the response to a linear resonator, we present the average number of photons $n_{\mathrm{lin}} = \frac{4\kappa_c}{(2\kappa_c + \kappa_i)^2}\frac{P_1}{hf_r}$ in a linear resonator[34,35] on top of Fig. 2a. The studied resonator has a constant transmission response for $n_{\mathrm{lin}} \ll 1$ with the Lorentzian lineshape, i.e. the response is identical to that of a linear resonator, and the linear resonator model works to describe the system in this regime where the resonator has only one photon at a time. For $n_{\mathrm{lin}} \gtrsim 1$, our resonator has already one photon in for significant amount of the time when a second photon tries to enter in. As the energy for the second photon differs significantly from the first photon - in the same way as the exited states

for a transmon qubit[36] - the second photon cannot interact and get transmitted through our resonator and the observed bleaching effect takes place, in line with the electrically induced transparency experiments[23,24]. The bleaching effect is illustrated with the energy diagram in Fig. 2b. The resonant input drive (cyan arrows) drives a transition between the photon states $|0\rangle$ and $|1\rangle$ but not between $|1\rangle$ and $|2\rangle$ due to energy mismatch. In Fig. 2c we further point out that the maximum power through the resonator is close to $P \approx \kappa_c h f_r$, which corresponds roughly of having one photon transmission at a rate of $\kappa_c$.

Next we measure the transition frequency $f_{12}$ for the transition $|1\rangle \leftrightarrow |2\rangle$ determining the Kerr nonlinearity for the second photon[26,27]. Towards this line, we perform a two-tone measurement presented in Fig. 2d. Here the transmission for the second tone with fixed power $P_2$ is measured as a function of its frequency $f_2$ and the power $P_1$ of the first tone with the frequency fixed to the $|0\rangle \leftrightarrow |1\rangle$ transition as $f_1 = f_{01} = 4.715$ GHz. Again, the second tone shows the same constant response independent of $P_1$ for $n_{\mathrm{lin}} \ll 1$ as the one tone measurement of Fig. 2a. For the higher powers with $n_{\mathrm{lin}} \gtrsim 1$, a second resonance mode at $f_2 = 4.44$ GHz appears. This corresponds to the $|1\rangle \leftrightarrow |2\rangle$ transition: once the $|1\rangle$ state has a considerable population and the bleaching takes place, the second process starting from this state is enabled. The transmission for the second tone can be therefore switched on and off with the first tone as presented in Fig. 2e. Without the first tone, the transmission is suppressed to $T < 2 \cdot 10^{-3}$. With the first tone switched on, the transmission increases by an order of magnitude. Figure 2f shows the output power of the second tone signal in resonance $f_2 = 4.44$ GHz as a function of $P_1$ and $P_2$. The response is linear for both

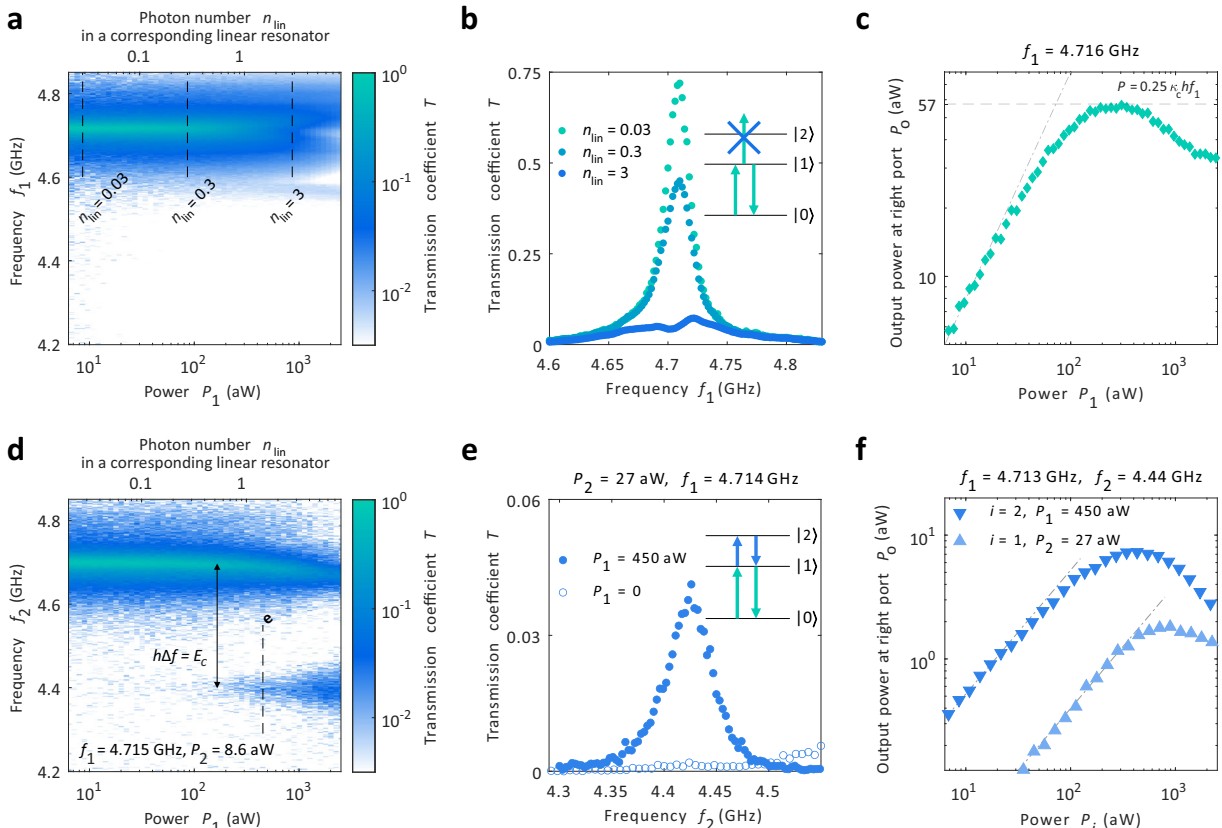

**Fig. 2 | Two-photon nonlinear effects. a** The transmission coefficient $T$ measured as a function of drive frequency $f_1$ and drive power $P_1$. The top scale of the figure shows the average photon number $n_{lin}$ of a corresponding linear resonator. **b** Line-cuts measured at the dashed lines of Fig. 2a. The energy diagram of the inset shows how the photon energy $hf_1$ (cyan arrows) does not match with the $|1\rangle \leftrightarrow |2\rangle$ transition. **c** Transmitted output power $P_O$ as a function of input power $P_1$ on resonance $f_1 = 4.716$ GHz. The dotted dashed line indicates the linear response $P_O = 0.7 P_i$ at low power and the dashed line the saturation to $P_O = 0.25\,\kappa_c h f_1$. **d** Two-tone measurement as a function of the pump power $P_1$ of the first tone and frequency $f_2$ of the probe. The first tone $f_1 = 4.715$ GHz is fixed to the resonance frequency and the probe power to $P_2 = 8.6$ aW. **e** Line-cut of Fig. 2d and the corresponding measurement with $P_1 = 0$. Now the transition $|1\rangle \leftrightarrow |2\rangle$ is visible if the first transition $|0\rangle \leftrightarrow |1\rangle$ is pumped with $P_1 \neq 0$ as indicated by the energy diagram. See Supplementary Note 1 for the reflection measurements corresponding to panels a-b and d–e. See Supplementary Information. **f** The output power at the second probe tone frequency as a function for the two powers $P_1$ and $P_2$. The dash dotted lines indicate the linear response at low power.

powers for $P_i < 100$ aW. At high power, the second photon transition also saturates and the transmitted power reduces.

The single JJ resonator has the Hamiltonian of a Cooper pair box which results in the transition frequencies as $hf_{01} = \sqrt{8 E_J E_c} - E_c$ and $hf_{12} = \sqrt{8 E_J E_c} - 2 E_c$ for $E_c/E_J \ll 1$, see ref. 36. Here $E_J$ is the Josephson energy set by the tunnel coupling of the junctions and $E_c = e^2/(2 C_\Sigma)$ the charging energy. The inductance for the first photon transition is correspondingly $L = \hbar^2/(4 e^2 E_J)$. The frequency difference of the first and second photon is simply $\Delta f = f_{01} - f_{12} = E_c/h$. The data of Fig. 2d yields then $E_c/h = 290$ MHz corresponding to the total capacitance of $C_\Sigma = 67$ fF. This value is close to the estimated junction capacitance of 56 fF based on the total area of $0.83\,\mu m^2$ and a typical capacitance of $68\,fF/\mu m^2$ of JJs[37]. Thus the JJs are the predominant source of capacitance and together with the Josephson coupling define the resonator mode. Using the measured transition frequency of $f_{01} = 4.715$ GHz yields further $E_J/h = 10.8$ GHz and the characteristic impedance $Z_r = \sqrt{L/C_\Sigma} = \frac{h}{e^2}\sqrt{E_c/2 E_J} = 0.5$ kΩ, comparable to the typical high-impedance resonators[5,7,9,28], hence showing the high impedance nature of the resonator. The value of the coupling $\kappa_c$ gives further the corresponding capacitance as $C_c = \sqrt{\kappa_c/8\pi^3 f_{01}^3 Z_0 Z_r} = 11$ fF with the input line impedance $Z_0 = 50\,\Omega$[28]. This value matches well the difference of $C_\Sigma$ and the estimated junction capacitance.

Figure 3 presents the resonator response in a high power regime. In the two-tone measurement of Fig. 3a we observe that the main resonator mode shifts towards lower frequencies similar to the high power response of JJ arrays[32]. In addition, the second photon mode at $f = 4.44$ GHz splits into two separate modes[38]. These features are due to the bare photon states $|n\rangle$ hybridizing to dressed states producing the so-called Autler-Townes effect as studied in refs. 27,39–41. In the Autler-Townes effect, a strong oscillating electrical field induces changes to the absorption and emission spectra. It is also known as the ac Stark shift. In the strong drive limit, the second tone transmission reaches up to a value of $T = 0.17$ at $P_1 = 100$ fW. In Fig. 3b and c, we present the transmission frequency of the probe tone as a function of the pump and probe frequencies $f_1$ and $f_2$. These measurements constitute energy diagrams of the resonator system as the frequencies corresponds directly the photon energies $hf_1$ and $hf_2$. With the lower $P_1$ of Fig. 3b, we see the bleaching effect of the $|0\rangle \leftrightarrow |1\rangle$ transition (Fig. 2a–c) at $f_1 = f_2 = 4.8$ GHz. The second photon transition $|1\rangle \leftrightarrow |2\rangle$ (Fig. 2d–f) is present at $f_1 = 4.8$ GHz, $f_2 = 4.5$ GHz and we see that it requires both of the frequencies to be set to these specific values resonantly. Interestingly driving the second photon transition with the stronger $P_1$ signal with $f_1 = 4.5$ GHz induces also a weaker bleaching effect to the first photon transition at $f_2 = 4.8$ GHz. This bleaching feature is slanted in the frequency map and connects towards the two-photon response along the line indicated by the gray arrow. Along this line, the energy conservation law $hf_1 + hf_2 = E_2 - E_0$ is followed. Here the

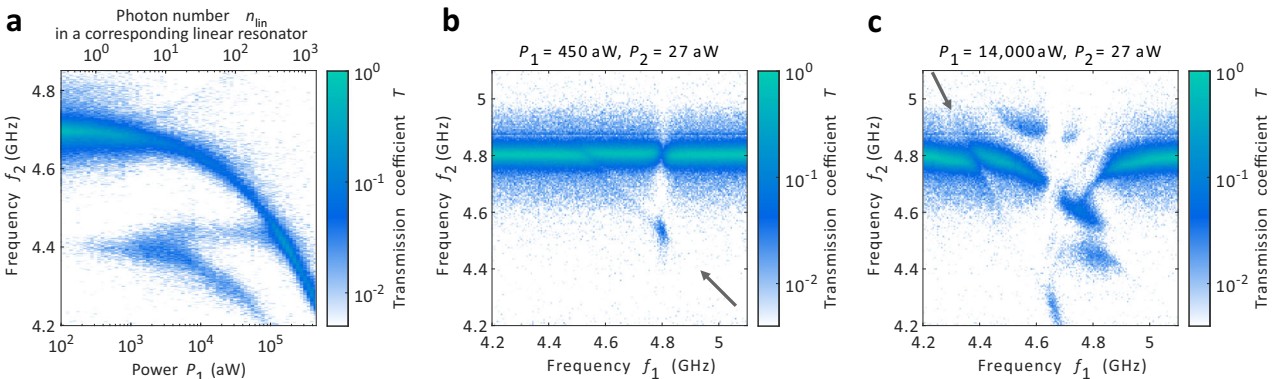

**Fig. 3 | Strong drive response and photonic energy diagrams. a** The two-tone measurement of Fig. 2d measured for two orders of magnitude stronger drive powers $P_1$. **b** The second tone transmission coefficient $T$ measured as a function of the two drive frequencies $f_1$ and $f_2$ at fixed $P_1 = 450$ aW and $P_2 = 27$ aW. **c** The energy diagram measurement of Fig. 3b repeated for a stronger drive power of $P_1 = 14,000$ aW.

$E_n = \sqrt{8E_J E_c}(n+1/2) - E_c(2n^2 + 2n + 1)/4$ are the eigenenergies of the $|n\rangle$ states[36]. Therefore the faint features along this line are signs of a two-photon process where one photon from each source leads to a transition directly from $|0\rangle$ to $|2\rangle$ for which the above energy conservation is valid.

In the higher power measurement of Fig. 3c several additional highly non-trivial features appear. We observe that the system has now many different combinations of $f_1$ and $f_2$ that lead to resonant transmission or suppression of the transmission. In this case, the dressed states are very likely needed to identify the relevant energies, which is beyond the scope of this study. However, we also see signs of the bare $|n\rangle$ states. In particular, we have strong features along the energy conservation line $2hf_1 + hf_2 = E_3 - E_0$ indicated by the gray arrow. Here we have a bleaching effect at $f_1 = 4.4$ GHz, $f_2 = 4.8$ GHz corresponding to drive at $2hf_1 = E_3 - E_1$ and probe at $hf_2 = E_1 - E_0$, and a three photon transition at $f_1 = 4.7$ GHz, $f_2 = 4.3$ GHz corresponding to drive at $2hf_1 = E_2 - E_0$ and probe at $hf_2 = E_3 - E_2$. These are analogous to the features in Fig. 3b but with two photon excitation processes. All of the processes here are such that they either start or finish with a direct single probe photon transition. Our interpretation is that these processes are the visible ones as processes involving multiple probe photons would need two or more probe photons to be present simultaneously in the input signal. This takes place only very rarely for the weak input tone we use. With the same intuition, the lower pump power case of Fig. 3b shows only single pump photon processes. For the higher pump power case of Fig. 3c, it becomes more likely to find two photons from the pump input signal simultaneously and therefore the two photon transitions $2hf_1 = E_2 - E_0$ and $2hf_1 = E_3 - E_1$ become visible. The findings of Fig. 3b–c thus demonstrate how this energy diagram technique enables to identify and analyse the different optics processes in the highly nonlinear resonators. Interesting and straightforward follow-up studies will be to replace the two-photon $2hf_1$ drive with one-photon drive with twice as high frequency and see if any selection rules apply to the different drive choices, or to increase the probe power to see if the transitions along the energy conservation lines become more prominent also outside of the above resonance conditions when the likelihood of obtaining photons from both signals simultaneously increases.

In summary, we have realized a high impedance resonator in the single Josephson junction limit exhibiting strong two-photon non-linear effects. The resonator is equivalent to a transmon qubit and hence our study shows experimentally that the qubit can be used - and perceived - as a resonator. An interesting future direction is to see if Kerr nonlinearity can be made comparable to the resonance frequency by increasing the charging energy $E_c$. Such a device would constitute a resonator where a single Cooper pair tunneling forms the photonic resonance mode and further increases the resonator impedance. The single JJ resonators demonstrated here constitute hence an elemental building block to build nonlinear quantum optics circuits and also to test the limits of the resonator impedance in the microwave domain.

## Methods

### Device fabrication

The resonator is made on top of an intrinsic silicon chip that has 200 nm of $SiO_2$ layer on top of it. First, rectangular contact pads and alignment marks were made with optical lithography. The contact pads are visible in Fig. 1a in the input lines and at the edge of the bottom ground plane in the middle of the figure. These structures are 5 nm / 45 nm Ti/Au metal films deposited with an e-beam evaporator. Next a 100 nm Nb layer was deposited in a liftoff process to define the ground below and above the resonator and the input lines up to the contact pad. As a last step, the JJ resonator and the interdigitized input coupler fingers were made with a standard Fulton-Dolan shadow evaporation technique. Electron beam lithography was used for defining the pattern in a PMMA-MMA resist stack. Then a 30 nm thick aluminum layer was evaporated at an angle of $-20°$. Then the tunnel barriers were defined by applying 3.2 mbar of oxygen for 10 min. As a last step another 40 nm thick aluminum film was deposited at an angle of $+20°$ to complete the SQUID structure of Fig. 1b. To obtain strong enough transmission and reflection response with the resonator, we made the capacitive couplers large enough so that $\kappa_c > \kappa_i$. The internal losses $\kappa_i$ of our device are larger than typically obtained for state-of-the-art superconducting devices[3,42–45]. Non-idealities in the device fabrication such as losses at the surfaces[43,44,46] are likely to give rise to these increased losses as we have not optimized the device fabrication process in the same way as done for the low loss devices. In addition, our ground planes are not perforated to pin vortices to the holes[46,47]. This may give rise to increased fluctuations of the magnetic flux $\Phi$ in the SQUID loop resulting in additional broadening of the resonance peaks.

### Measurements

The measurements were performed in a dilution refrigerator at a base temperature of 10 mK. The measurement setup was identical to ref. 48, with the addition of a splitter between the 20 dB attenuator and the cryostat input. The pump RF generator for the two-tone measurements was connected to the added splitter input such that the two tones were combined to the signal going into the cryostat. Also, between the splitter and the pump RF generator, a 20 dB attenuator was used to attenuate reflected signals from the generators, suppressing spurious interference effects. The internal clock of the pump generator was kept unsynchronized in the presented measurements.

We also made control experiments with synchronized tone signals. The synchronized signals made changes to the response only when the two drive frequencies were equal. All the other features were identical with and without synchronization, and hence the results of the study remain the same also with the syncronized input tones. During the single tone measurements, the pump was switched off, but the setup remained the same. The outgoing signal from the right port (Fig. 1a) was always measured, which means that for the transmission measurements the in-going signal was through the left port and for reflection measurements the in-going signal was through the right port.

In order to characterize the resonator and tune the resonance frequency, a magnetic field was applied perpendicular to the surface of the chip. The SQUID structure was chosen to allow for this tuneability, helping to identify the resonator mode thanks to its flux periodic response. To this end, the transmitted signal was measured while sweeping the magnetic field in order to find the resonance mode of the device (Fig. 1d). This measurement was made with constant power from the generator. The means that the power at the resonator input varies over the different frequencies, as the cable attenuation is frequency dependent. This does not however affect the main result of the figure, since the main resonance mode is clearly visible over the range of frequencies shown. In order to find the transmission coefficient and to determine the powers going into and out of the resonator, a calibration was made to determine the attenuation and amplification of the input and output lines. The method for this was identical to ref. 48. The calibration was done at 4.7 GHz and all powers are determined with this calibration constant. The calibration was also repeated at the second photon transition frequency of 4.4 GHz, but since this one differed by less than 10% from the 4.7 GHz calibration, the overall results remain unchanged even if this calibration would be used for the lower frequencies instead. The difference between the calibrations is also within the uncertainty of the calibration, motivating further to disregard it. See Supplementary Note 2 for the calibration data. See Supplementary Information.

From the measurement of Fig. 1d, the operation point at $\Phi = 0.4\Phi_0$ was chosen. This operation point was selected because: 1) It has a strong response for the resonator mode without the avoided crossing-like features present e.g. at $f = 6.2$ GHz. 2) The spurious background signals, both for magnetic flux independent contributions and magnetic flux dependent contributions[49] are low. Our interpretation of the additional features is the following: The device has an extra flux independent mode at $f = 6.2$ GHz, likely from an additional two-level system in the junction oxide[49]. The resonator hybridizes with this mode if tuned to this frequency. The flux dependent other modes, fainter than the main mode, arise likely from the chosen SQUID structure as their flux periodicity is comparable to (but not exactly the same as) the resonance mode. Determining the precise origin of these modes would require further efforts beyond the scope of this study, for example by studying if these features depend on the size of the Josephson junctions or the size of the shift of the shadows in the SQUID loop resulting in from the shadow evaporation technique. The flux independent response that oscillates in frequency at $f = 5.8$–5.9 GHz and $f = 7.4$–8.0 GHz on the other hand results in likely from a spurious coupling directly from the input line to the output line as it does not depend on the flux and does not shift the main mode response in frequency or flux.

After choosing the flux operation point, the couplings were determined by fitting a Lorentzian to the transmission coefficient, assuming $\kappa_L = \kappa_R$ since the device design is symmetric. The operation point was later changed to $\Phi = -1.4\Phi_0$ for the measurements of Figs. 2 and 3 due to the spurious background transmission being lower at this operation point for the two-tone measurements. The resonator had otherwise identical properties (couplings, internal losses and resonance frequency) at this operation point. For the data in Fig. 3b–c, the magnetic field was tuned so that the resonance frequency is higher by 100 MHz as compared to Figs. 2 and 3a. This tuning was done to have the full response fitting in the 4.2–5.1 GHz window that was clean from spurious background signals present for $f < 4.2$ GHz, see Fig. 1d.

## Data availability

Source data are provided with this paper. Other data that support the findings of this study are available from the corresponding authors upon request. Source data are provided with this paper.

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

## Acknowledgements

We thank the Knut and Alice Wallenberg Foundation through the Wallenberg Center for Quantum Technology (WACQT), the European Union (ERC, QPHOTON, 101087343), Swedish Research Council (Dnr 2019-04111) and NanoLund for financial support. Views and opinions expressed are however those of the author(s) only and do not necessarily reflect those of the European Union or the European Research Council Executive Agency. Neither the European Union nor the granting authority can be held responsible for them.

## Author contributions

V.F.M. conceived the presented idea. S.A., H.H. and S.H. fabricated the devices with support from A.R. and V.F.M. S.A. performed the measurements and analysed the data with contributions from all the other coauthors. S.A. and V.F.M. prepared the manuscript with input from all the other coauthors. V.F.M. supervised the project.

## Funding

## Competing interests

The authors declare no competing interests.
