## [Transparent Peer Review file · Nature Communications]

High-Impedance Microwave Resonators with Two-Photon Nonlinear Effects

Corresponding Author: Professor Ville Maisi

Version 0:

Reviewer comments:

Reviewer #1

(Remarks to the Author)

This paper examines a non-linear high-impedance (~500 ohm) microwave resonator formed by a single SQUID. Its behavior is examined function of power, and while driving with a pump and probe from two signal sources. The experiments and results are clearly explained, appear technically sound, and the content is broadly useful to the superconducting electronics and quantum information community.

I consider this paper suitable for publication, with a few recommendations:

One piece of data I would like to see, but realize may be missing given that transmission and reflection are measured independently per the supplementary information, is how the reflected signal changes while bleaching or saturation occurs. For example, the reflected signal power would be very interesting to see included in Figs. 2b and 2c, to see how much of the signal is getting reflected vs eaten up through internal losses. I'm not so set on this that I would require it to meet publication, but if the data is available it should certainly be presented.

The internal Q of the device is not very high ($2 * \pi * f_{01} / \kappa_{i} = 1620$), I'd like to see some commentary on this (is this typical for the fab process used, is there some source of noise in the system, do we even care?).

As a minor point, in addition to the JJ capacitance and two coupling capacitances C_c , C_{Σ} also includes the node (teal region of Fig. 1a) capacitance to ground. This may be rather small in this device.

Reviewer #2

(Remarks to the Author)

The authors have implemented a highly nonlinear microwave resonator based on a Josephson junction circuit. Specifically, they have developed a high impedance resonator exhibiting strong two-photon nonlinear effects. This work also provides a method to characterize the nonlinearity using a two-tone measurement scheme.

I find this work both interesting and useful for the scientific community engaged in microwave quantum engineering, and I recommend it for publication.

I do have a few questions for the authors:

1)
In Figure 2b, the evolution of the resonance peak shows a decrease as the power of the input field increases. At high power, the average photon number is 3. This seems counterintuitive: if the system is strongly nonlinear, increasing the power while pumping at a fixed frequency should, in theory, cause the average photon number to saturate at 1, as the "second photon" cannot enter the cavity. Can the authors provide a more detailed explanation of this phenomenon?

2)
Figure 3 presents very interesting results, especially Figure 3c, which contains a wealth of information. The authors explicitly state that "identifying the relevant energies... is beyond the scope of this study." However, I believe it would be beneficial to

elaborate on this part. For example, is there an intuitive reason why the processes $2f_1 + f_2$ are visible while other processes are not?

Reviewer #3

(Remarks to the Author)
Please see the attached file.

Reviewer #4

(Remarks to the Author)
I co-reviewed this manuscript with one of the reviewers who provided the listed reports as part of the Nature Communications initiative to facilitate training in peer review and appropriate recognition for co-reviewers.

Version 1:

Reviewer comments:

Reviewer #1

(Remarks to the Author)
My concerns have been addressed in the latest revision. However, I recommend replacing the white-teal-blue color scale in Figs. S1a and S1c with the same white-blue-teal scale used in the plots of the main manuscript. Thank you.

Reviewer #3

(Remarks to the Author)
Please check the attachment for full review.

Reviewer #4

(Remarks to the Author)
I co-reviewed this manuscript with one of the reviewers who provided the listed reports. This is part of the Nature Communications initiative to facilitate training in peer review and to provide appropriate recognition for Early Career Researchers who co-review manuscripts.

In the manuscript “High-Impedance Microwave Resonators with Two-Photon Nonlinear Effects”, the authors demonstrate measurements on a two-Josephson Junction based device with direct capacitance coupling to a transmission line. Using single- and two-tone spectroscopy measurements, the authors demonstrate resonator-like behavior in the low power regime and non-linear behavior in the high-power regime. The results are competently presented and mostly well-explained. However, the major claims made in the paper has conceptual flaws and the connection between these claims and the measured results is not well-established. Additionally, the experimental methods employed in the study, including single and two-tone spectroscopy, are standard and have been widely utilized in the field of superconducting circuits for over a decade. These techniques are well-established, as evidenced by earlier works like those of I. Schuster et al. (Nature Physics volume 4, pages 382–385 (2008)) and A. Wallraff et al. (Phys. Rev. Lett. 99, 050501 (2007)). Given the absence of novel physical phenomena or innovative device designs and measurement methodologies, I cannot recommend publication in Nature Communications.

Detailed comments:

The major claim in the paper "*By taking a Josephson Junction array based high impedance resonator to the limit of consisting effectively only of one junction, results in strong non-linear effects already for the second photon while maintaining a high impedance of the resonance mode.*" is conceptually misleading. A resonator is characterized by its linear response to external drives. By reducing the Josephson junction array to a single junction, the non-linearity of the device is significantly enhanced, as the measurements in the paper confirmed. This non-linearity is tied to the system Hamiltonian itself, instead of photon number in the mode. Although the physical phenomenon behind this description is perceivable, i.e. the 0-1 photon transition frequency is very different than the 1-2 photon transition frequency, this effect is essentially the characteristic of a standard transmon device rather than a resonator anymore. Although there exist intermediate regimes where the non-linearity of a resonance mode is comparable to the photon loss rate, in which many novel physical phenomena can be explored (Andersen, C. K., *PRA* (2020). Yamaji, T., et al *PRA* (2022)), the parameters presented in this paper (anharmonicity $E_c/h = 290$ MHz, loss rate $(k_c+k_i)/2\pi=15$ MHz) falls rather in the “transmon” regime, whose “few photon” dynamic has been well-studied in the superconducting qubit community (Schuster, D. Yale University (2007), Koch, J., et al. *PRA*(2007), Lescanne, R, et al. *Phys. Rev. Applied* (2019).) The paper could be much stronger if the authors could demonstrate a more compelling application for their design. Additionally, including one or two paragraphs on the motivation behind the research would strengthen the article.

The second half of the statement "*The experiment yields thus resonators with the strong interactions both between individual resonator photons and from the resonator photons to other electric quantum systems.*" is not fully substantiated by presented data, as only a single transmon mode was measured.

A DC SQUID-like loop was used in the experiment for the flux tuneability of the effective inductance of the device. However, the authors didn't explain why the specific flux bias point was chosen in the experiment and the necessity for having such a loop instead of a single junction. Regarding the flux sweep presented in Figure 1d on the SQUID device, the readers would benefit from additional explanations about the other features in the sweep. Specifically, the modulation of other modes and the anti-crossing behavior of the SQUID with these modes should be addressed. Providing more context and interpretation of these features would enhance the understanding of the results. Furthermore, for Figure 3, which discusses the strong drive response, the energy curve of the transmon-like device under a high power regime can probably be explained by the Stark shift or other effects. If this performance is worth presenting, it would be beneficial to predict and analyze it using physical expressions to provide a more thorough understanding.

Overall, the study effectively presents continuous-wave measurements on a device akin to a standard transmon and provides reasonable interpretations of observed photon blockade and multi-wave mixing effects. However, these effects, typical of transmons, alongside the unjustified portrayal of the device as a high-impedance resonator, fall short of the standard for publication in Nature Communications. The paper would benefit from a reevaluation of its conceptual framework and a clearer demonstration of novelty or significant advancements in the field.

Dear Authors,

Thank you for your detailed responses and the revisions made to your manuscript. I appreciate the effort you've put into addressing the points I raised. Overall, I am satisfied with the changes, particularly regarding the clarifications on the technical aspects of the work.

That said, I still have some doubts about whether this work should be published in *Nature Communications*. Especially, regarding whether the non-linearity of the device should be tied to the Hamiltonian or the photon number. I apologize if my point in the previous report was unclear. The main concern I would like to raise is as follows: for any device whose Hamiltonian has a Taylor expansion with lowest-order terms that are quadratic in charge and flux, at low drive power, the device will always behave like a resonator. This is because the non-linear effects are associated with higher-order terms, which only become observable under strong driving. Therefore, I don't think experimentally demonstrating this behavior is necessarily a novel insight, particularly given that the device used in the paper has already been fully demonstrated in previous work.

If we follow your definition of "few-photon" non-linearity in resonators, then the well-established superconducting qubit—transmon, which operates in the same parameter regime as the device presented in this paper—would also qualify as a non-linear "high-impedance" resonator. Then, the claim regarding the innovation in this work, particularly the statement: "*So far, the few-photon nonlinearities have remained untapped for high-impedance systems, despite the potential to build non-linear quantum optics devices with unprecedentedly strong and fast interaction dynamics,*" does not hold up. The few-photon behavior of the transmon has been extensively studied for more than a decade. For example, the transmon has been shown to operate as a qutrit (Bianchetti et al., PRL 105. 223601 (2010)), a qudit (Wu et al., PRL 125, 170502 (2020)), and most recently, up to the 12th level (Wang et al., arXiv:2407.17407). The driven dynamics of these systems, from zero to few-photon levels and beyond, have also been thoroughly explored (Peterer et al., PRL 114. 010501 (2015); Lescanne et al., PRA 11. 014030 (2019)). Furthermore, the majority of these studies involve time-domain dynamics, which are notably absent from the current work.

Regardless, in your response, you stated:

"Our system can therefore be perceived both in the transmon picture as the referee thinks, but also as a non-linear resonator too. The latter seems to have caught the Referee by surprise. Thus we find that this novelty in the thinking is already sufficient to credit the publication of the work. We are also sure that the Referee is not the only one who realizes with this work that this simple system can indeed be perceived as the most non-linear resonator possible. With that, our study shows a significant change to the usual way of thinking."

While I agree that offering new perspectives on established concepts can be valuable, I do not believe that the worthiness of a paper for publication should be determined by a novel redefinition of a known concept alone. In my opinion, the strength of a research paper lies in its ability to address significant problems or uncover new phenomena that advance the field. Publications like I. C. Rodrigues et al. (*Nature Communications*, 10, 5359 (2019)) or Bishop et al. (*Nature Physics*, 5, 105-109, 2009) exemplify research that either solves pressing issues or identifies new phenomena

critical to our understanding of quantum devices.

As I mentioned in my initial review, my surprise is not due to your novel interpretation, but rather stems from these key questions:

1. Is there any physics problem that cannot be solved without this paper's idea of resonators with two-photon nonlinear effects?
2. Has any new phenomenon been discovered in this paper that is essential to our current understanding?

Please forgive my ignorance if the authors believe these questions are trivial or self-evident. However, I believe that directly addressing these concerns, and potentially providing applications, would not only clarify my doubts but also help readers understand the broader implications of this research and open avenues for further exploration.

Response to the review reports

S. Andersson,¹ H. Havir ¹, A. Ranni,¹ S. Haldar ¹ and V. F. Maisi ¹

¹*NanoLund and Solid State Physics, Lund University, Box 118, 22100 Lund, Sweden*

(Dated: September 11, 2024)

I. RESPONSE TO REVIEW REPORT #1

Referee: This paper examines a non-linear high-impedance (~ 500 ohm) microwave resonator formed by a single SQUID. Its behavior is examined function of power, and while driving with a pump and probe from two signal sources. The experiments and results are clearly explained, appear technically sound, and the content is broadly useful to the superconducting electronics and quantum information community.

I consider this paper suitable for publication, with a few recommendations:

Our response: Thank you for the positive assessment. We address below each of the recommendations and have updated the manuscript according to the recommendations.

Referee: One piece of data I would like to see, but realize may be missing given that transmission and reflection are measured independently per the supplementary information, is how the reflected signal changes while bleaching or saturation occurs. For example, the reflected signal power would be very interesting to see included in Figs. 2b and 2c, to see how much of the signal is getting reflected vs eaten up through internal losses. I'm not so set on this that I would require it to meet publication, but if the data is available it should certainly be presented.

Our response: This is certainly an interesting aspect. As part of the initial characterization of the device, we measured the reflection coefficient response but with a different set point for the flux ($\Phi = 0.4 \Phi_0$) than in Fig. 2 of the main article. With this set point, the resonance frequency is 150 MHz higher. The ranges for the frequency and power were also somewhat different than in Fig. 2. Since we focused in this work to use the transmission response to study the non-linear properties of the resonator, we deemed the repetition of the reflection coefficient measurements precisely at the same operation point/parameter range as used in Fig. 2 not crucial when performing the final set of measurements.

To give the readers a chance to consider the reflection coefficient response, we prepared now a supplemental material presenting the reflection coefficient data. We prefer to have the data available in a supplemental material. If the data would be included in Fig. 2, the differences in the flux setting and ranges is probably likely to cause unnecessary confusion for the non-expert readers as they would need to consider the details of the experiment to understand the differences in the data. We also added the data used for the reflection and transmission coefficient calibration into the supplemental material as that is also based on measuring the reflection coefficient.

Referee: The internal Q of the device is not very high ($2\pi f_{01}/\kappa_i = 1620$), I'd like to see some commentary on this (is this typical for the fab process used, is there some source of noise in the system, do we even care?).

Our response: Our internal losses are indeed much larger than what is obtained for the best performing superconducting devices where Q values exceeding 10^6 are reported. The fabrication process is definitely one likely reason for this. We have not spend such an extensive effort to minimize the losses arising from surface losses as for example the large superconducting quantum computing groups have successfully made.

Another viable reason for our higher losses is that we have not made vortex traps to the ground planes as is done e.g. in PRL 106, 120501 (2011) by having holes in the ground planes. Therefore our device is more prone to moving vortices in the ground planes which may increase fluctuations in the SQUID flux setting and thus broaden resonance peaks.

The internal losses are important especially in terms of designing the device so that the resonator response is well visible in the experiment. We made one iteration round in the device fabrication to increase the coupler capacitance to get $\kappa_c > \kappa_i$. Having κ_c larger than the internal losses is important since if the internal losses dominate, the resonator response is suppressed in that case.

We bring up these relevant points in the methods section of the updated manuscript with the added text *"To obtain strong enough transmission and reflection response with the resonator, we made the capacitive couplers large enough so*

that $\kappa_c > \kappa_i$. The internal losses κ_i of our device are larger than typically obtained for state-of-the-art superconducting devices [3, 42–45]. Non-idealities in the device fabrication such as losses at the surfaces [43, 44, 46] are likely to give rise to these increased losses as we have not optimized the device fabrication process in the same way as done for the low loss devices. In addition, our ground planes are not perforated to pin vortices to the holes [46,47]. This may give rise to increased fluctuations of the magnetic flux Φ in the SQUID loop resulting in additional broadening of the resonance peaks.”

Referee: As a minor point, in addition to the JJ capacitance and two coupling capacitances C_c , C_Σ also includes the node (teal region of Fig. 1a) capacitance to ground. This may be rather small in this device.

Our response: This is true. The device has also capacitance to ground. We mention this now with the added text “Here, the total capacitance C_Σ contains the JJ capacitance, capacitance from the resonator middle line (shown in teal) to ground, and two input capacitances C_c which connect the resonator to input and output lines (shown in blue).” in the updated manuscript main text where the different capacitances are introduced.

II. RESPONSE TO REVIEW REPORT #2

Referee: The authors have implemented a highly nonlinear microwave resonator based on a Josephson junction circuit. Specifically, they have developed a high impedance resonator exhibiting strong two-photon nonlinear effects. This work also provides a method to characterize the nonlinearity using a two-tone measurement scheme.

I find this work both interesting and useful for the scientific community engaged in microwave quantum engineering, and I recommend it for publication.

Our response: Thank you! We address the point-by-point questions below.

Referee: I do have a few questions for the authors:

1) In Figure 2b, the evolution of the resonance peak shows a decrease as the power of the input field increases. At high power, the average photon number is 3. This seems counterintuitive: if the system is strongly nonlinear, increasing the power while pumping at a fixed frequency should, in theory, cause the average photon number to saturate at 1, as the “second photon” cannot enter the cavity. Can the authors provide a more detailed explanation of this phenomenon?

Our response: We see that our choice of the figure labels causes unnecessary confusion here. The photon numbers indicated with $\langle n \rangle$ in the initial submission were for a linear resonator that has the same low power resonance frequency and couplings as our non-linear resonator. We described this in detail in the main text but the figure notation evidently does not carry through this important point. To remove this confusion we have now changed the variable name to “ n_{lin} ”. We also have now added “Photon number n_{lin} in a corresponding linear resonator” to the top of Fig. 2a and 2d. Hopefully this clarifies the figure notation so that this confusion is avoided now.

The reason why we want to use the photon number in a linear resonator is that it provides precisely the intuition that the referee describes: As seen in Fig. 2a, when $n_{\text{lin}} \ll 1$, our non-linear resonator behaves exactly the same way as the linear resonator: We have a Lorentzian transmission response independent on the input power. When the photon number of the linear resonator approaches or exceeds unity ($n_{\text{lin}} = 0.3 - 3$), our non-linear resonator saturates as the “second photon” cannot enter into the cavity, precisely as the referee points out.

Referee: 2) Figure 3 presents very interesting results, especially Figure 3c, which contains a wealth of information. The authors explicitly state that “identifying the relevant energies... is beyond the scope of this study.” However, I believe it would be beneficial to elaborate on this part. For example, is there an intuitive reason why the processes $2f_1 + f_2$ are visible while other processes are not?

Our response: We are delighted to see that the referee shares the same excitement about these results as we do. Our intuition is that it is easier to find two photons from the strong pump tone for the $2f_1$ component than for example two photons from the weak probe to obtain a $2f_2$ component. We provide now this as an interpretation in the updated manuscript with the added text “All of the processes here are such that they either start or finish with a direct single probe photon transition. Our interpretation is that these processes are the visible ones as processes involving

multiple probe photons would need two or more probe photons to be present simultaneously in the input signal. This takes place only very rarely for the weak input tone we use. With the same intuition, the lower pump power case of Fig. 3b shows only single pump photon processes. For the higher pump power case of Fig. 3c, it becomes more likely to find two photons from the pump input signal simultaneously and therefore the two photon transitions $2hf_1 = E_2 - E_0$ and $2hf_1 = E_3 - E_1$ become visible.”

III. RESPONSE TO REVIEW REPORT #3

Referee: In the manuscript “High-Impedance Microwave Resonators with Two-Photon Nonlinear Effects”, the authors demonstrate measurements on a two-Josephson Junction based device with direct capacitance coupling to a transmission line. Using single- and two-tone spectroscopy measurements, the authors demonstrate resonator-like behavior in the low power regime and non-linear behavior in the high-power regime. The results are competently presented and mostly well-explained. However, the major claims made in the paper has conceptual flaws and the connection between these claims and the measured results is not well-established. Additionally, the experimental methods employed in the study, including single and two-tone spectroscopy, are standard and have been widely utilized in the field of superconducting circuits for over a decade. These techniques are well-established, as evidenced by earlier works like those of I. Schuster et al. (Nature Physics volume 4, pages 382–385 (2008)) and A. Wallraff et al. (Phys. Rev. Lett. 99, 050501 (2007)). Given the absence of novel physical phenomena or innovative device designs and measurement methodologies, I cannot recommend publication in Nature Communications.

Our response: We agree with the referee that the measurement methods used are indeed widely used in the community. We do not claim this to be the novelty of our work, so there is no need to consider the technical aspects of the measurements for that matter.

The novelty of our work is making a Josephson junction -based resonator in the limit of having effectively only a single Josephson junction. This yields the strongest possible non-linearities into these kind of resonators. The other two referees, together with us authors, judge this to be suitable for Nature Communications. Referee #3 opposes this here and below with the argument that our manuscript would have conceptual flaws such that the system cannot be considered as a resonator. As we show below in the response, the made argumentation is not correct. We find all the properties present in our experiment/device which the Referee lists as missing items that would be needed for a proper resonator. It seems that perceiving this transmon-based system as a resonator (that for instance accepts only one photon at the fundamental resonance frequency drive) is novel to the Referee. We hope that our response is clear enough to realize this point. We are surely happy to elaborate more if anything still remains unclear for the Referee.

Referee: Detailed comments: The major claim in the paper ” By taking a Josephson Junction array based high impedance resonator to the limit of consisting effectively only of one junction, results in strong non-linear effects already for the second photon while maintaining a high impedance of the resonance mode.” is conceptually misleading. A resonator is characterized by its linear response to external drives. By reducing the Josephson junction array to a single junction, the non-linearity of the device is significantly enhanced, as the measurements in the paper confirmed. This non- linearity is tied to the system Hamiltonian itself, instead of photon number in the mode. Although the physical phenomenon behind this description is perceivable, i.e. the 0-1 photon transition frequency is very different than the 1-2 photon transition frequency, this effect is essentially the characteristic of a standard transmon device rather than a resonator anymore. Although there exist intermediate regimes where the non-linearity of a resonance mode is comparable to the photon loss rate, in which many novel physical phenomena can be explored (Andersen, C. K., PRA (2020). Yamaji, T., et al PRA (2022)), the parameters presented in this paper (anharmonicity $E_C/h = 290$ MHz, loss rate $(kc+ki)/2\pi=15$ MHz) falls rather in the “transmon” regime, whose “few photon” dynamic has been well-studied in the superconducting qubit community (Schuster, D. Yale University (2007), Koch, J., et al. PRA(2007), Lescanne, R, et al. Phys. Rev. Applied (2019).) The paper could be much stronger if the authors could demonstrate a more compelling application for their design. Additionally, including one or two paragraphs on the motivation behind the research would strengthen the article.

Our response: We disagree here with the referee:

1. Our single junction resonator has a well established linear response regime at low power. Figure 1e shows this response which is the ordinary Lorentzian transmission and reflection response. Figure 2a shows further that the transmission response is constant for low power as for a linear resonator (i.e. the transmitted power is

directly proportional to the input power with the linear response Lorentzian lineshape). Figure 2a shows also further that at high power the response changes from this, i.e. the non-linearities set in at high enough photon number. For our resonator that just happens already for average photon numbers below one. Therefore, unlike the referee states, the non-linearity is in fact tied to the photon number in the mode also for our resonator. For low enough photon number $n \ll 1$, the resonator response is linear and for high power $n \gtrsim 1$, the non-linearities become relevant.

2. The referee considers also that there would be an exclusive difference with the situation where the non-linearity is tied to system Hamiltonian itself vs. if it is tied to the photon number in the mode. Such an exclusive difference does not exist for any resonator. Even for a very weakly non-linear resonator, the non-linearity is tied to system Hamiltonian itself. Similarly, the non-linearity is tied to the photon number in the mode, not only for the weakly non-linear resonator but also for our strongly non-linear resonator. So both properties are valid for all of the non-linear resonators. To demonstrate these cases in a simple manner, let us consider the resonator Hamiltonian

$$\hat{H} = \sum_n \hbar\omega_n |n\rangle \langle n|, \quad (1)$$

where $|n\rangle$ is n photon state with eigenenergy $\hbar\omega_n$. The non-linearity is now tied to this system Hamiltonian itself as it is determined by how much the eigenenergy spacing ($\hbar\omega_{n+1} - \hbar\omega_n$) vary as a function of n . Reducing this variance reduces the non-linearity. The reduction applies all the way to the limit that all of the level spacings are equal, i.e. have the constant value ($\hbar\omega_{n+1} - \hbar\omega_n$) = $\hbar\omega_r$ with the eigenenergies $\hbar\omega_n = \hbar\omega_r n$. Now we have a fully linear resonator with

$$\hat{H} = \sum_n \hbar\omega_n |n\rangle \langle n| = \hbar\omega_r \sum_n n |n\rangle \langle n| = \hbar\omega_r \hat{a}^\dagger \hat{a}. \quad (2)$$

Therefore, for any strength of the non-linearity (i.e. for any choice of the $\hbar\omega_n$ values), the non-linearity is determined by the system Hamiltonian, independent of the input/output couplings. This applies also for the fully linear resonator with vanishing non-linearity.

Next, considering the second "choice", i.e. the question if the non-linearity is tied to the photon number or not. Here, the simplest counter argument is based on considering very weak drive. For a sufficiently low drive amplitude the photon number stays within $n = 0$ and $n = 1$ for any resonator, even for the linear one. In this regime, only the energies $\hbar\omega_1$ and $\hbar\omega_0$ are relevant, and the resonator response depends only on the energy difference of these states, independent if the resonator is linear or non-linear. Most importantly the resonance frequency is $\omega_1 - \omega_0$ in this linear response regime. Only if the photon number is increased above this value, the non-linear effects set in if there are any non-linearities in the system Hamiltonian. Therefore, the non-linearity is tied to the photon number for every choice of the $\hbar\omega_n$ eigenenergy spectrum.

With this, we have demonstrated that both of the options that the referee thinks as exclusive, apply for our resonator. In addition they both also apply for any other more weakly non-linear resonator, independent of the input coupling.

Because of these misconceptions that the Referee had when preparing their report, the Referee thought that our statements are conceptually misleading. As the argumentation made in the Referee report is not correct, there is in fact nothing conceptually misleading in our statements. Our system can therefore be perceived both in the transmon picture as the referee thinks, but also as a non-linear resonator too. The latter seems to have caught the Referee by surprise. Thus we find that this novelty in the thinking is already sufficient to credit the publication of the work. We are also sure that the Referee is not the only one who realizes with this work that this simple system can indeed be perceived as the most non-linear resonator possible. With that, our study shows a significant change to the usual way of thinking.

Referee: The second half of the statement "The experiment yields thus resonators with the strong interactions both between individual resonator photons and from the resonator photons to other electric quantum systems." is not fully substantiated by presented data, as only a single transmon mode was measured.

Our response: This is true. Thanks a lot for pointing out the inaccuracy in this statement. We indeed do not show the strong coupling of the resonator to another electric quantum system. Coupling Josephson junction resonators with high impedance has been already done by several works. See e.g. A. Stockklauser et al. Phys. Rev. X **7**, 011030

(2017), A. Landig et al., Nature Comm. **10**, 5037 (2019), P. Scarlino et al., Nature Comm. **10**, 3011 (2019), M. Scigliuzzo et al., Phys. Rev. X **12**, 031036 (2022) and A. Ranni et al., arXiv:2308.14887 (2023).

We changed the sentence to read now *"Our experiment yields thus resonators with the strong interactions between individual resonator photons, complementing the earlier JJ resonator works [5, 13, 29–31] coupling the resonator photons strongly to other electric quantum systems."*, so that it is evident that the first part is shown in our study and the second one is established by other works in the field, not by the present study.

Referee: A DC SQUID-like loop was used in the experiment for the flux tuneability of the effective inductance of the device. However, the authors didn't explain why the specific flux bias point was chosen in the experiment and the necessity for having such a loop instead of a single junction.

Our response: The specific flux bias point was chosen so that there's minimal amount of any spurious background transmission in the response as seen in Fig 1d. The main reason for choosing the SQUID-structure is that it makes it easier to identify the right mode from the flux periodic response and tune the resonance frequency to the 4 -8 GHz measurement band. We provide these details now in the methods section with the added text *"The SQUID structure was chosen to allow for this tuneability, helping to identify the resonator mode thanks to its flux periodic response"* and *"This operation point was selected because: 1) It has a strong response for the resonator mode without the avoided crossing-like features present e.g. at $f = 6.2$ GHz. 2) The spurious background signals, both for magnetic flux independent contributions and magnetic flux dependent contributions are low."*

Referee: Regarding the flux sweep presented in Figure 1d on the SQUID device, the readers would benefit from additional explanations about the other features in the sweep. Specifically, the modulation of other modes and the anti-crossing behavior of the SQUID with these modes should be addressed. Providing more context and interpretation of these features would enhance the understanding of the results.

Our response: This is a good point. We now comment on the extra features in the methods section with *"Our interpretation of the additional features is the following: The device has an extra flux independent mode at $f = 6.2$ GHz, likely from an additional two-level system in the junction oxide [49]. The resonator hybridizes with this mode if tuned to this frequency. The flux dependent other modes, fainter than the main mode, arise likely from the chosen SQUID structure as their flux periodicity is comparable to (but not exactly the same as) the resonance mode. Determining the precise origin of these modes would require further efforts beyond the scope of this study, for example by studying if these features depend on the size of the Josephson junctions or the size of the shift of the shadows in the SQUID loop resulting in from the shadow evaporation technique. The flux independent response that oscillate in frequency at $f = 5.8 - 5.9$ GHz and $f = 7.4 - 8.0$ GHz on the other hand results in likely from a spurious coupling directly from the input line to the output line as it does not depend on the flux and does not shift the main mode response in frequency or flux."*

Referee: Furthermore, for Figure 3, which discusses the strong drive response, the energy curve of the transmon-like device under a high power regime can probably be explained by the Stark shift or other effects. If this performance is worth presenting, it would be beneficial to predict and analyze it using physical expressions to provide a more thorough understanding.

Our response: The strong drive can indeed be seen analogous to an ac-Stark shift. In other words that the strong oscillating electrical field from the drive changes the Hamiltonian and hence the eigenenergies of the system. We already commented this in the submitted version of the manuscript with the text and the references: *"These features are due to the bare photon states $|n\rangle$ hybridizing to dressed states producing the so-called Autler-Townes effect as studied in Refs. 27, 39, 40."* We added now *"In the Autler-Townes effect, a strong oscillating electrical field induces changes to the absorption and emission spectra. It is also known as the ac Stark shift."* so that the readers understand that the Autler-Townes effect is the same as the ac Stark shift. Note that we added also Ref. P. Y. Wen et al., Phys. Rev. Lett. **120**, 063603 (2018), to the above sentence as we missed this work discussing the Autler-Townes effect in the first submission.

Regarding the comment "If this performance is worth presenting, it would be beneficial to predict and analyze it using physical expressions to provide a more thorough understanding", we are not really sure what the intention of the Referee is here. We interpret it as a suggestion to extend the study towards analysing the strong drive regime more in detail. As these are not essential for the claims made in the manuscript, we prefer to leave this as a follow-up study and refrain from extending the manuscript too much. If there is a specific missing item for the presented main

results, we would ask the referee to provide the detailed information about that and we are surely happy to address and correct that into the manuscript.

Referee: Overall, the study effectively presents continuous-wave measurements on a device akin to a standard transmon and provides reasonable interpretations of observed photon blockade and multi-wave mixing effects. However, these effects, typical of transmons, alongside the unjustified portrayal of the device as a high-impedance resonator, fall short of the standard for publication in Nature Communications. The paper would benefit from a reevaluation of its conceptual framework and a clearer demonstration of novelty or significant advancements in the field.

Our response: As argued above, we disagree with this assessment that the device cannot be perceived as a resonator. Please also note the other two Referee reports that find the resonator view valid and also find our work fulfilling the high standards for publishing in Nature Communications.

Response to the review reports

S. Andersson,¹ H. Havir ¹ A. Ranni,¹ S. Haldar ¹ and V. F. Maisi ¹

¹*NanoLund and Solid State Physics, Lund University, Box 118, 22100 Lund, Sweden*

(Dated: December 2, 2024)

I. REVIEWER #1 (REMARKS TO THE AUTHOR):

Referee: My concerns have been addressed in the latest revision. However, I recommend replacing the white-teal-blue color scale in Figs. S1a and S1c with the same white-blue-teal scale used in the plots of the main manuscript. Thank you.

Our response: Very good point. We changed now the color scale in Figs. S1a and S1c to be the same as the main manuscript to maintain the consistency.

II. REVIEWER #2 (REMARKS TO THE AUTHOR):

No further remarks sent.

III. REVIEWER #3 (REMARKS TO THE AUTHOR):

Referee: Dear Authors,

Thank you for your detailed responses and the revisions made to your manuscript. I appreciate the effort you've put into addressing the points I raised. Overall, I am satisfied with the changes, particularly regarding the clarifications on the technical aspects of the work. That said, I still have some doubts about whether this work should be published in Nature Communications. Especially, regarding whether the non-linearity of the device should be tied to the Hamiltonian or the photon number. I apologize if my point in the previous report was unclear. The main concern I would like to raise is as follows: for any device whose Hamiltonian has a Taylor expansion with lowest-order terms that are quadratic in charge and flux, at low drive power, the device will always behave like a resonator. This is because the non-linear effects are associated with higher-order terms, which only become observable under strong driving. Therefore, I don't think experimentally demonstrating this behavior is necessarily a novel insight, particularly given that the device used in the paper has already been fully demonstrated in previous work.

If we follow your definition of "few-photon" non-linearity in resonators, then the well-established superconducting qubit—transmon, which operates in the same parameter regime as the device presented in this paper—would also qualify as a non-linear "high-impedance" resonator. Then, the claim regarding the innovation in this work, particularly the statement: "So far, the few-photon non-linearities have remained untapped for high-impedance systems, despite the potential to build non-linear quantum optics devices with unprecedentedly strong and fast interaction dynamics," does not hold up. The few-photon behavior of the transmon has been extensively studied for more than a decade. For example, the transmon has been shown to operate as a qutrit (Bianchetti et al., PRL 105. 223601 (2010)), a qudit (Wu et al., PRL 125, 170502 (2020)), and most recently, up to the 12th level (Wang et al., arXiv:2407.17407). The driven dynamics of these systems, from zero to few-photon levels and beyond, have also been thoroughly explored (Peterer et al., PRL 114. 010501 (2015); Lescanne et al., PRA 11. 014030 (2019)). Furthermore, the majority of these studies involve time-domain dynamics, which are notably absent from the current work.

Regardless, in your response, you stated:

"Our system can therefore be perceived both in the transmon picture as the referee thinks, but also as a non-linear resonator too. The latter seems to have caught the Referee by surprise. Thus we find that this novelty in the thinking is already sufficient to credit the publication of the work. We are also sure that the Referee is not the only one who realizes with this work that this simple system can indeed be perceived as the most non-linear resonator possible. With that, our study shows a significant change to the usual way of thinking."

While I agree that offering new perspectives on established concepts can be valuable, I do not believe that the worthiness of a paper for publication should be determined by a novel redefinition of a known concept alone. In my opinion, the strength of a research paper lies in its ability to address significant problems or uncover new phenomena

that advance the field. Publications like I. C. Rodrigues et al. (Nature Communications, 10, 5359 (2019)) or Bishop et al. (Nature Physics, 5, 105-109, 2009) exemplify research that either solves pressing issues or identifies new phenomena critical to our understanding of quantum devices.

As I mentioned in my initial review, my surprise is not due to your novel interpretation, but rather stems from these key questions:

1. Is there any physics problem that cannot be solved without this paper's idea of resonators with two-photon nonlinear effects?
2. Has any new phenomenon been discovered in this paper that is essential to our current understanding?

Please forgive my ignorance if the authors believe these questions are trivial or self-evident. However, I believe that directly addressing these concerns, and potentially providing applications, would not only clarify my doubts but also help readers understand the broader implications of this research and open avenues for further exploration.

Our response: We also wish to thank the Referee for their time and effort to provide useful comments and criticisms to improve the manuscript content and clarity.

We also appreciate and agree with Referee's ambition to make a real change to the scientific community and ultimately to the society by either providing new tools to overcome existing problems or make a significant contribution to the current understanding. This impact can, however, be determined ultimately only in the future when we see which new directions and outcomes a given study has opened up. Our view is that the energy diagram technique presented in this work revealing the multi-photon processes along the energy conserving lines could be such a new tool. These results are also not presented in any of the earlier works, including the ones above that the Referee is referring to.

The above works pointed out by the Referee use an ancilla resonator for the photonic modes. For example, the Nature Physics, 5, 105-109, (2009) publication uses a superconducting transmon to induce non-linearities to an ordinary linear resonator. Our work deviates considerably from this by using the transmon structure directly as the resonator without any other ancilla systems that span an unnecessarily complex Hilbert space for the photon modes. In the first review round, Referee's standpoint was that these kind of resonators are not possible. With their statements "*the major claims made in the paper has conceptual flaws and the connection between these claims and the measured results is not well-established*" and "*the unjustified portrayal of the device as a high-impedance resonator*", it is rather evident to our mind that before this work, the Referee was considering that such highly non-linear resonators do not exist. We are happy to see that - based on this review - these issues are now cleared and the Referee also sees the system now as a proper resonator. To our mind, bringing such a new conceptual way to the thinking is sufficient merit for publishing the work. The other two referees also found the work worth publishing, so our view is that this question is now settled.

IV. REVIEWER #4 (REMARKS TO THE AUTHOR):

Referee: I co-reviewed this manuscript with one of the reviewers who provided the listed reports. This is part of the Nature Communications initiative to facilitate training in peer review and to provide appropriate recognition for Early Career Researchers who co-review manuscripts.

V. ADDITIONAL CHANGES MADE TO CORRECT TYPOGRAPHICAL ERRORS

- "Non-linear" replaced with "nonlinear".
- Several typographical mistakes were corrected as highlighted in blue in the updated manuscript.
- A factor of " $(n + 1/2)$ " was missing in the equation " $E_n = \sqrt{8E_J E_c}(n + 1/2) - E_c(2n^2 + 2n + 1)/4$ " on page 4, so we added this factor. This equation can be found from Ref. 36 as Eq. (2.11) that we cited already in the initial version. The other term, " $-E_J$ ", in Eq. (2.11) of the Ref. 36 is not relevant as it is a constant that just shifts the overall energy reference level. Therefore it can be neglected (or added if so preferred). The $(n + 1/2)$ factor on the other hand is very central as it describes how the total energy E_n increases linearly for increasing photon number n . This is the linear contribution to the resonator total energy.
- Ref. 13 is updated to the published article (Phys. Rev. Res. **6**, 043134 (2024)).